# A Microenvironment-Related Nine-Gene Signature May Predict Survival in Mycosis Fungoides Patients at Diagnosis

**DOI:** 10.3390/cells12151944

**Published:** 2023-07-27

**Authors:** Silvia Alberti-Violetti, Maria Rosaria Sapienza, Marcello Del Corvo, Federica Melle, Giovanna Motta, Luigia Venegoni, Lorenzo Cerroni, Carlo Cota, Alessandro Pileri, Emilio Berti, Stefano A. Pileri

**Affiliations:** 1Dermatology Unit, Fondazione IRCCS Ca’ Granda Ospedale Maggiore Policlinico, 20122 Milan, Italy; emilio.berti@gmail.com; 2Department of Pathophysiology and Transplantation, University of Milan, 20122 Milan, Italy; luigia.venegoni@unimi.it; 3Division of Diagnostic Haematopathology, European Institute of Oncology IRCCS, 20139 Milan, Italy; mariarosaria.sapienza@ieo.it (M.R.S.); marcello.delcorvo@gmail.com (M.D.C.); federica.melle@ieo.it (F.M.); giovanna.motta@ieo.it (G.M.); stefano.pileri@unibo.it (S.A.P.); 4Research Unit of Dermatopathology, Medical University of Graz, 8036 Graz, Austria; lorenzo.cerroni@medunigraz.at; 5Dermatopathology Laboratory San Gallicano, Dermatological Institute IRCCS, 00144 Rome, Italy; carlocota@yahoo.it; 6Dermatology Unit, IRCCS AUBO, 40138 Bologna, Italy; alessandro.pileri2@unibo.it; 7Department of Medical and Surgical Sciences, Alma Mater Studiorum University of Bologna, 40138 Bologna, Italy

**Keywords:** cutaneous lymphoma, mycosis fungoides, microenvironment, prognosis, gene expression profiling

## Abstract

Mycosis fungoides (MF) is the most common cutaneous lymphoma characterized by an indolent course. Prognosis is stage-based but this approach does not reflect the different outcomes within stages. Considering that tumor microenvironment is known to be involved in MF pathogenesis and progression, we decided to investigate 99 MF cases by using the PanCancer Immune Profiling Panel. We identified and validated a signature of 9 genes able to predict MF survival and distinguish a high-risk group with a worse outcome from a low-risk group of cases with a better outcome. At the molecular level, low-risk vs. high-risk cases reported a global upregulation of immune genes, enriched in cytokines, and a higher density of dendritic cells and mast cells, possibly associated with a more favorable clinical course.

## 1. Introduction

Mycosis fungoides (MF) is the most common primary cutaneous T-cell lymphoma (CTCL), accounting for around 50% of all cutaneous lymphoma [1]. MF is a rare disease with a poorly understood pathobiology, multiple clinical presentations, and outcomes [2].

The majority of MF patients present at the early stage disease, characterized by localized (IA) or diffuse patches and plaques (IB), with no evidence of high tumor burden in the blood or systemic involvement. The early stage MF has an indolent course with a good prognosis (median survival > 15 years, 5-year survival > 80%) [1]. However, 25–33% of these patients can progress to an advanced stage, completely changing the prognosis. Around one-third of all MF patients present with advanced-stage disease, characterized by tumor (IIB), or erythroderma (III), or blood involvement (IVA1), nodal involvement (IVA2) or visceral and/or bone marrow involvement (IVB). During progression to advanced stages, the MF course becomes aggressive, reporting 5-year overall survival (OS) rates of 62% to 23–15%, even if a minority can survive for longer than 5 years [3,4]

Today, prognosis and clinical management are still stage-based, in spite of the above-mentioned wide range of outcomes within stages. These differences in survival have created the need to identify new markers useful for predicting the progression and clinical outcome of MF patients already diagnosed, and to eventually help clinicians to adopt the most suitable therapeutic approach [3,5].

In the last few years, many studies have investigated the molecular steps behind the development and progression of CTCL, which includes recurrent deletions of tumor suppressors (e.g., *ARID1A*, *CDKN2A/B*, *PTPRC*, *SOCS1*, and *STK11*), gene mutations of *PLCG1* and *JAK3*, and epigenetic alterations such as histone/chromatin modifications and hypermethylation of multiple tumor suppressor genes (e.g., *BCL7a* and *CDKN2A/B*). Like in other CTCL, aberrations in JAK-STAT signaling seem to be pivotal in MF, causing a pro-tumorigenic inflammatory environment. [6]. In particular, studies about CTCL tumor microenvironment have shown that not only the tumoral cells but also the inflammatory and resident skin cells (such as keratinocytes and cells of the innate and adaptive immune system) play a determinant role in MF evolution [7].

In this work, we analyzed 99 MF cases by using the PanCancer Immune Profiling Panel (NanoString Technologies), specifically designed to interrogate 730 immune-targeting genes and detect genes, immune cell types, and molecular pathways that possibly involved in the tumor immune microenvironment (TIME) remodeling.

Thanks to this approach, we identified a nine-gene, TIME-derived signature, capable of predicting patient outcome at diagnosis, with relevant clinical and therapeutic implications.

## 2. Materials and Methods

### 2.1. Patient Selection

In this study, at diagnosis, we collected 99 formalin-fixed, paraffin-embedded (FFPE) skin biopsies of MF cases, staged according to the 2007 MF and SS staging revision. [8]. Forty-three out of ninety-nine cases were selected from the database of the Dermatology Unit, Fondazione IRCCS Ca’ Granda—Ospedale Maggiore Policlinico, Milan (Italy) and were used as discovery set. This set included 18 females and 25 males with a median age of 62 (15–91); 22 were in the early stage (1 IA, 10 IB, 11 IIA) and 21 in the advanced stage (13 IIB, 5 III, 2 IVA1, 1 IVA2). The remaining 56 MF cases were collected from three independent tertiary care centers in Graz (Austria), Bologna (Italy), and Rome (Italy) and were included in the validation set. The patients in this set were 20 females and 36 males with a median age of 63 (14–87); 17 of them were in the early stage (4 IA, 9 IB, 4 IIA), 39 in the advanced stage (29 IIB, 3 III, 5 IVA1, 2 IVA2). 

The current study was conducted in accordance with the Declaration of Helsinki.

### 2.2. Total RNA Extraction and NanoString nCounter Assay

Total RNA was extracted from 3 sections of 10-μm of each FFPE tissue sample using the RNAstorm FFPE extraction kit (Cell Data Sciences, Fremont, CA, USA). All the samples passed the quality control step with 260/280 and 260/230 ratios ≥ 1.8 and RNA concentration ≥ 60 ng/µL. 

Gene expression analysis was measured via the NanoString nCounter Analysis System (NanoString Technologies, Seattle, WA, USA) with nCounter PanCancer Immune Profiling Panel that performs multiplex gene expression analysis, including 730 immune-related target genes, 40 housekeeping genes, and additional positive and negative genes. The panel includes genes from different immune cell types (B-cells, T-cells, T-helper 1, T-helper 2, T regulatory, cytotoxic cells, dendritic cells, macrophages, mast cells, neutrophils and Natural Killer cells), common checkpoint inhibitors, cancer testis antigens, and genes covering both the adaptive and innate immune response. The nCounter CodeSet containing capture and reporter probes was hybridized to 300 ng of total RNA for 20 hours at 65 °C, according to the manufacturer’s instructions. Hybridized samples were loaded into the nCounter Prep Station for post-hybridization processing. Target mRNA was assessed with nCounter Digital Analyzer, using maximum scan resolution. The NanoString system computes the relative abundance of each mRNA transcript of interest, through a multiplexed hybridization assay and digital readouts of fluorescent barcoded probes that are hybridized to each transcript.

### 2.3. Gene Expression Profiling and Cell Type Profiling of the Discovery Set and the Validation Set

The raw data file from the nCounter Digital Analyzer was analyzed via the NanoString nCounter nSolver™ 4.0.70 using the NanoString Advanced Analysis Module 2.0 plugin. The Advanced Analysis Module 2.0 software was used for quality control (QC), normalization, differential expression analysis, and cell-type profiling with default parameters. The estimated amount of immune cell types in each sample was calculated via NanoString nSolver cell type abundance score.

Genes were recognized as differentially expressed (DEGs) if reporting a log2FC ≥ 0.5 and *p* value ≤ 0.01. The identified differentially expressed genes were next analyzed via pathway analysis using Enrichr bioinformatic tool and KEGG_2021_Human gene set library [9].

### 2.4. Statistical Survival Analysis 

All the statistical analyses were conducted within the statistical environment R v4.2.2 https://www.r-project.org/ (accessed on 20 July 2023). Association of OS and gene expression was investigated via univariate Cox regression analysis using the survival package. Unlike the Kaplan–Meier method, Cox proportional hazards regression is able to provide an effect estimate by quantifying the difference in survival between patient cohorts but it does not require the assumption of survival distributions. The hazard ratio (HR) obtained from this analysis was used to identify candidate genes associated with the OS. Genes with HR < 1 were considered as protective genes, meaning that a lower expression is associated with poor prognosis while genes with HR > 1 were defined as risky genes, where poor prognosis is related to higher expression. A *p* value of <0.05, resulting from log-rank test, was considered statistically significant for the selection of the OS-related genes.

### 2.5. Gene Expression Signature-Based Prognostic Risk Score

Significant genes resulting from Cox regression analysis were used to create a prognostic scoring system based on this formula:$$\sum_{i=1}^{n.genes}\left(gene\;i\;Cox\;regression\;coefficient\right)x\;\left(gene\;i\;expression\;level\right)$$

The patients were ranked by their prognostic scores. An optimum threshold on the score to separate patients into high-risk and low-risk groups was determined using the MaxStat package [10] to identify the cutoff value producing the maximum log-rank score in the discovery set. Kaplan–Meier plots were constructed, and a long-rank test was implemented to evaluate differences in OS for MF patients. 

### 2.6. Development and Validation of Nomogram

We constructed a nomogram of 3-year and 5-year OS to visualize the results of the Cox regression analysis for both discovery and validation sets with the R package, ‘rms’. The concordance index (C-index), which shows the consistency between predicted probability and observed outcome, can be used to estimate the predictive performance of nomogram. The C-index value ranges from 0.5 to 1.0, where 0.5 indicates random and 1.0 represents a perfect match. A higher C-index indicates a better consistency between the prediction and the observed result. When C-index reaches a value of at least 0.7, the nomogram prediction can be considered significant.

## 3. Results

### 3.1. Development of a Nine-Gene Prognostic Signature Able to Distinguish High-Risk vs. Low-Risk MF Patients

First, we analyzed the discovery set in order to identify molecular markers able to discriminate patients with different outcomes.

A univariate Cox proportional hazard regression analysis of all 730 genes was performed to determine the association between gene expression and OS of MF patients.

This analysis identified a nine-gene signature whose lower expression was significantly associated with poor prognosis (LogRank < 0.05, Table 1). Next, a prognostic score was used to estimate a patient’s risk of death and was determined by the linear combination of logarithmically transformed gene expression levels weighted by average Cox regression coefficient. Hence, the prognostic scores were assigned for all MF patients and the cohort was then divided into two groups, namely low- and high-risk groups accordingly. Kaplan–Meier analysis was performed, and a log-rank test was used to determine significant differences in OS among the two groups. High-risk patients were found to have a significantly worse OS compared to low-risk patients (Figure 1A).

The nine-gene prognostic signature was tested in the validation set. Patients were ranked based on their score and divided into high-risk and low-risk groups. Kaplan–Meier analysis confirmed a significant difference among the two cohorts, as reported in the discovery set (*p* value < 0.001). (Figure 1B).

Hence, to evaluate the robustness of our signature we divided early-stage cases and advanced-stage cases from both the discovery and validation sets, and we analyzed them using the nine-gene prognostic signature. As shown in Figure 1C,D, the nine-gene prognostic signature significantly recognized high-risk and low-risk patients also in early or advanced MF groups.

### 3.2. The Low-Risk vs. High-Risk MF Patients Presented a Tumor Immune Microenvironment (TIME) with a Prominent Activation of Cytokine Signaling Pathway and Enrichment of Dendritic Cells and Mast Cells

We conducted a differential expression analysis of low-risk vs. high-risk discovery set patients and found 42 DEGs, all up-regulated but one, supporting a global up-regulation of immune response in low-risk MF patients with a more favorable outcome (Appendix A). Next, we interrogated the identified 42 DEGs by functional analysis and found that the cytokine–cytokine receptor interaction was the pathway most significantly activated in low-risk MF cases (Figure 2A). We also interrogated the validation set and found 162 DEGs, 134 up- (82.7%) and 28 down-regulated genes (17.3%) (Appendix A). The cytokine–cytokine receptor interaction resulted to be the biological pathway mostly represented in low-risk vs. high-risk cases, confirming the discovery set findings. (Figure 2B).

Next, to assess if specific immune cell types of TIME may be associated with the clinical outcome of MF patients, we performed the cell-type profiling of low-risk vs. high-risk MF cases of the discovery set. We detected a higher infiltration of dendritic cells (DCs), CD8 T-cells and mast cells (MCs) in low-risk vs. high-risk cases. (*p* value ≤ 0.05 via the Mann–Whitney Test). We also conducted the cell type profiling analysis in the validation set, confirming that low-risk cases are significantly enriched in DCs and MCs, possibly involved in the inflammatory, cytokine-mediated response (Figure 2C,D).

## 4. Discussion

In this study, we developed a nine-gene prognostic signature able to discriminate at diagnosis MF patients with a better or worse clinical outcome, namely low-risk and high-risk MF patients, regardless of their clinical stage. 

Compared to other nodal lymphomas, a standardized prognostic index for MF is still not available, because of its rarity and heterogeneity. Stage remains the strongest prognostic factor, though other clinical and histological markers have been validated, such as age, gender, folliculotropism, and CD30 expression, but none of them are able to predict which early-stage patients can evolve to late stages [2,3,11] In recent years, numerous genomic alterations have been found with a potential diagnostic and prognostic purpose. It is well-known that genomic instability is typical of advanced stages with aggressive course [3]. Litvinov et al. [12] stratified patients with CTCL into three prognostic groups, based on targeted gene expression profiles. Authors were able to identify genes more frequently expressed in a favorable disease course (such as *WIF1*) compared to those in patients with a poor prognosis (such as *IL17F*). Shin et al. [13] grouped CTCL patients based on gene expression, regardless of the stage, finding that the cluster with worse prognosis and therapy-resistant disease showed an up-regulation of genes involved in lymphocyte activation and inflammation. MiR-155 and miR-200 b were also significantly associated with patient overall survival and were considered valid predictors for patient clinical outcomes [14].

Recently, the role of the TIME has been found to be critical in regulating tumor cell migration and proliferation and influencing disease progression, opening up the possibility of using new therapeutic approaches targeting the non-tumoral components of the microenvironment. 

In view of these findings, we decided to analyze, at the transcriptional level the TIME of MF, in search of immune molecular markers capable of predicting the disease progression and clinical outcome of MF patients, at diagnosis. 

Thanks to our multi-stage bioinformatics approach, we have developed a nine-gene prognostic signature capable of recognizing high- and low-risk MF cases at diagnosis. 

As reported by functional enrichment analysis, low-risk cases have been found to have a more inflamed TIME, predominantly enriched in cytokines, known to regulate the migration and recruitment of immune cells into and out of the tissue and to guide the spatial organization and cellular interactions of immune cells within tissues. As is already known, the cytokines may exert opposite functions by triggering an anti-tumor immune response in some contexts or, alternatively, may contribute to chronic inflammation and immune escape inducing tumor growth [15]. The nine-gene prognostic signature included different genes encoding for cytokines or chemokines involved in the recruitment of inflammatory or cancer-associated cells and reported as dysregulated in hematological and/or solid cancers. *CCL13* and *CCL23* are chemoattractant for various cells of the immune systems, including DCs and MCs [16,17,18]. CCL23 has been also reported to participate in the recruitment of cancer-related cells and its decrease is associated with shortened patient survival in solid tumors [18]. CCL18 is chemo-attractive for T cells, B cells, and immature DCs. Its expression is reported in MF, but it has not been determined whether CCL18 has a positive or negative effect on the development of CTCL in vivo [19]. 

In our study, cell type profiling analysis also confirmed that the skin microenvironment of low-risk MF has a higher density of different immune cell types, such as DCs and MCs, compared to high-risk MF. 

MCs do not have a well-known role in tumor growth. In solid tumors as well as hematological neoplasms, a high count of MCs has been correlated with both progression and good prognosis [20]. Regarding CTCL, MCs are significantly increased in CTCL lesions compared to normal skin but data about their role are few and contradictory. Rabenhorst et al. [21] demonstrated a pro-tumorigenic activity and correlation with a higher microvessel density. On the other hand, Eder et al. [20] found a greater predominance of MCs in early stages compared to late-stage CTCL.

Regarding DCs, they are antigen-presenting cells which are involved in immune activation in inflammatory benign lesions as well as in malignant lymphoid proliferations. There are three types of cutaneous DCs: Langerhans cells, plasmacytoid DCs, and dermal DCs, which represent the majority of DCs in CTCL infiltrate [22]. In our study both MC and DCs associated with a better clinical course, but given their debating role in CTCL, further studies are needed to corroborate our findings. 

Although our TIME-based prognostic signature was validated in independent cohorts to predict tumor prognosis, this study has some limitations. Because of the retrospective design, the results were biased to some extent, and, despite the rarity of the disease, the number of samples is small. Thus, a well-designed, prospective, international, multicenter study is needed to verify our findings prospectively. A further contribution for better understanding the relationships between TIME and neoplastic cells may be provided by the new technologies allowing single-cell analysis in FFPE tissue sections. 

## 5. Conclusions

In conclusion, for the first time, we have found a TIME-related prognostic signature in MF patients. The results presented here are promising because they not only help to identify at diagnosis patients at high risk of progression, notwithstanding the clinical stage, but also point to new clinical applications for personalized therapy in patients with MF.

## Figures and Tables

**Figure 1 cells-12-01944-f001:**
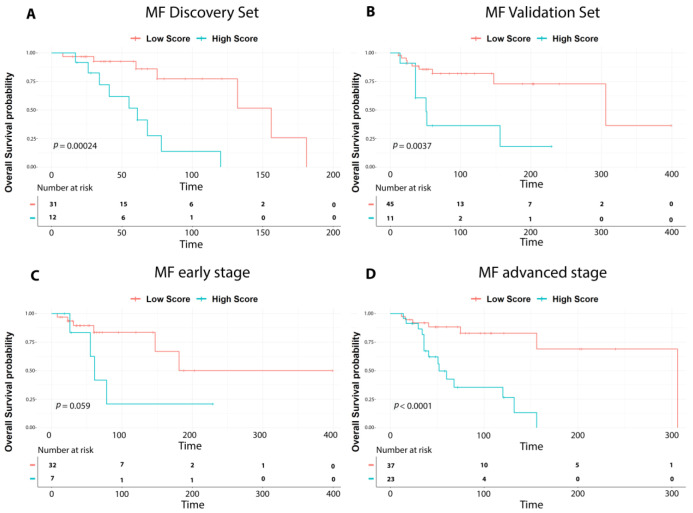
Survival analysis of MF cases. Overall survival curves according to the nine-gene prognostic signature showing significant differences in outcome between high-risk (high score in light blue) vs. low-risk cases (low score in red) in: (**A**) discovery set cases; (**B**) validation set cases; (**C**) early stage MF (IA-IIA); (**D**) advanced-stage MF (IIB-IVA2). *p*-values were calculated with the log-rank test.

**Figure 2 cells-12-01944-f002:**
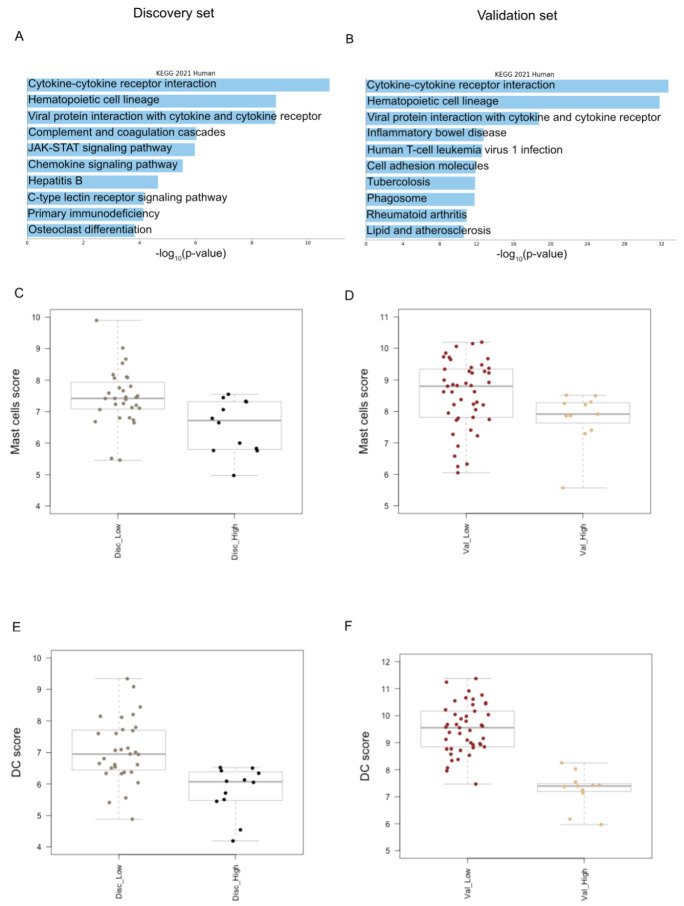
Tumor immune microenvironment analysis. (**A**) The bar chart displays the top 10 Kegg biological pathways significantly enriched in low-risk vs. high-risk MF cases of discovery set. The pathways, colored in light blue, are displayed based on the −log_10_ (*p* value), from the most enriched to the lowest one (*p* value adjusted <0.05); (**B**) the top 10 Kegg biological pathways significantly enriched in low-risk vs. high-risk MF cases of validation set; (**C**) box plots show a higher infiltration of mast cells in low-risk vs. high-risk cases in discovery set as well as in (**D**) validation set; (**E**) box plots illustrate the higher level of dendritic cells in low-risk vs. high-risk cases in discovery set, namely Disc_low and Disc_High and also in (**F**) validation set, namely Val_Low and Val_High.

**Table 1 cells-12-01944-t001:** List of 9 genes identified by Cox regression analysis. Based on their hazard ratio (<1), their lower expression is associated with poor prognosis. HR: hazard ratio.

Gene	LogRank	HR	Gene Class	Immune Response Category	Annotation
CCL13	0.0051	0.5856	Immune Response-Cell Type specific	Chemokines	Cell Type specific, Chemokines and receptors, Inflammatory response
CCL23	0.0113	0.7450	Immune Response	Chemokines, Regulation	Chemokines and receptors, Regulation of Inflammatory response
CCL26	0.0152	0.8097	Immune Response	Chemokines	Chemokines and receptors, Inflammatory response
CCL18	0.0158	0.8412	Immune Response	Chemokines	Chemokines and receptors, Anti-inflammatory cytokines
TRAP	0.0284	0.3845	Immune Response		Innate immune response
FCER2	0.0328	0.8044	Immune Response		Adaptative immune response, CD molecules, Inflammatory response
CCL24	0.0346	0.7381	Immune Response	Chemokines. Regulation	Chemokines and receptors, Regulation of Inflammatory response
CD209	0.0349	0.5643	Immune Response-Cell Type specific	Cell Functions	Basic cell functions, Cell Type specific, CD molecules
LILRB3	0.0370	0.7764	Immune Response	Regulation	CD molecules, Regulation of immune response

## Data Availability

All data that support the findings of this study are available from the corresponding author upon reasonable request.

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
