# Peer review of "A Microenvironment-Related Nine-Gene Signature May Predict Survival in Mycosis Fungoides Patients at Diagnosis"

_cells, 2023, doi:10.3390/cells12151944_

Round 1

Reviewer 1 Report

The manuscript addresses a very relevant and significant topic about the need for better prognostic biomarkers in CTCL. Given the rarity of the disease, more such studies are needed to improve the management of CTCL patients. Overall, the manuscript is well-structured, the figures are easy to understand and the methodology is described in sufficient detail. The authors have made adequate effort to cite relevant and recent publications from this field. 

For the above reasons, I'm recommending that this paper be accepted following minor revision outlined below: 

1) Although the authors mention in the introduction section that advanced stage comprises Stage IIB to IVB, they do not reclarify in the context of Fig. 1C and 1D, which stages they are defining as "Early-stage" versus "Advanced MF". It would make it easier for reader to reclarify this in the legend or text associated with Fig. 1C,D in the Results section.

2) Since the patient samples were primarily from Italy, it is possible that the gene signature is biased by ethnicity. I request authors to add a sentence in the discussion section to acknowledge the limitation of their patient sample type. Also, in the methods section, I would like the authors to provide additional detail on the patient sample, example: age at diagnosis, gender, body surface area covered by patches/plaques, blood involvement (measured by flow cyotometry). 

Author Response

The manuscript addresses a very relevant and significant topic about the need for better prognostic biomarkers in CTCL. Given the rarity of the disease, more such studies are needed to improve the management of CTCL patients. Overall, the manuscript is well-structured, the figures are easy to understand and the methodology is described in sufficient detail. The authors have made adequate effort to cite relevant and recent publications from this field.  For the above reasons, I'm recommending that this paper be accepted following minor revision outlined below: 

1) Although the authors mention in the introduction section that advanced stage comprises Stage IIB to IVB, they do not reclarify in the context of Fig. 1C and 1D, which stages they are defining as "Early-stage" versus "Advanced MF". It would make it easier for reader to reclarify this in the legend or text associated with Fig. 1C,D in the Results section.

We thank the reviewer for this suggestion. We have added the stages for early stage and advanced stage in the text of the figure 1 (line 177). Moreover, we have inserted these data in the patient descriptions (lines 72-78)

2) Since the patient samples were primarily from Italy, it is possible that the gene signature is biased by ethnicity. I request authors to add a sentence in the discussion section to acknowledge the limitation of their patient sample type. Also, in the methods section, I would like the authors to provide additional detail on the patient sample, example: age at diagnosis, gender, body surface area covered by patches/plaques, blood involvement (measured by flow cyotometry). 

We thank the reviewer for the suggestions. In both sets, patients are all Caucasians, but they are not only Italian because, in the validation set, there are also patients from a tertiary referral center in Austria. We have added more details about patients, such as age and gender. Unfortunately, we do not have data about mSWAT in all the patients because clinical data had been retrospectively collected. Similarly, we do not have data about flow cytometry in all the early stages because this is not required in routine real-world experience.

Reviewer 2 Report

Please add the nomogram about cox regression to further confirm the experimental results

Please adjust the size of the text in Figure 1

Please reproduce the results of the enrichment in Figure 2 A,B and add the figure notes.

Minor editing of English language required

Author Response

Please add the nomogram about cox regression to further confirm the experimental results .

Thank you for your suggestion. We have added nomograms for both discovery and validation cohorts, confirming the strong prediction performance of our signature. (paragraph 2.6, lines 133-141 of the result)

Please adjust the size of the text in Figure 1

Thank you for your suggestion. We have modified the size of the text in Fig 1

Please reproduce the results of the enrichment in Figure 2 A,B and add the figure notes.

Thank you for your suggestion. We have modified Figure 2 adding the explanation in the figure legend.

Minor editing of English language required.

Thank you for your suggestion, a minor editing of English has been performed by a native English speaker.

Reviewer 3 Report

Alberti-Violetti et al. present a microenvironment-related 9-gene signature that can be used to predict survival in mycosis fungoides patients. There are several points that need to be addressed to support their claims:

Major points:

1)      Can the authors elaborate on the different stages and molecular steps involved in MF in the introduction?

2)      Can the authors provide details on the 730 immune-target genes and molecular pathways that were analyzed?

3)      Provide the rationale for using Cox regression model and the rationale behind cut-off chosen for obtaining the 9-gene signature.

4)      Can the 9-gene signature be used to predict the stage of MF?

5)      Have the authors performed any experiments to validate the protein expression of the 9 genes and the correlation to MF?

6)      Can the authors assess other cell types involved in TME such as B cells, CAFs?

7)      What is the predictive power and accuracy score of the 9-gene signature? Do all the genes contribute equally to the model?

8)      Can the authors perform any additional experiments to validate their 9-gene signature model?

Minor points:

9)      In Figure 2A, add a scale bar and color key.

10)   Expand OS in line 36.

Author Response

Alberti-Violetti et al. present a microenvironment-related 9-gene signature that can be used to predict survival in mycosis fungoides patients. There are several points that need to be addressed to support their claims:

Major points:

1)     Can the authors elaborate on the different stages and molecular steps involved in MF in the introduction?

Thank you very much for your suggestion. We have improved the introduction, expanding the section about stages and molecular steps. We have also added reference 6.

2)      Can the authors provide details on the 730 immune-target genes and molecular pathways that were analyzed?

Thank you for your suggestion. We have summarily included, in the materials and methods section (lines 88-92), the classes of genes included but we also provide the complete list of genes of the panel in a separate table for supplementary information if the reviewer of the editor requests it.

3)      Provide the rationale for using Cox regression model and the rationale behind cut-off chosen for obtaining the 9-gene signature.

Thank you very much for your suggestion. We have decided to use Cox regression model because it is the most commonly used method for the analysis of survival data and it does not require the assumption of survival distributions. We explain this rationale better in paragraph 2.4 (lines 113 - 121)

As for the logic behind the choice of the cut-off for the selection of 9-gene signature, we used the log-rank test and set the significance level of p.value to 0.05, which is the reference value above which the null hypothesis is retained (see lines 106-107).

4)      Can the 9-gene signature be used to predict the stage of MF?

Thank you very much for your question. The answer is no, this signature is not predictive of the stage but, alternatively, it is predictive of the prognosis despite the stage, seeking to answer the question “which early stage can progress?”

5)      Have the authors performed any experiments to validate the protein expression of the 9 genes and the correlation to MF?

Thank you very much for your question. We are conducting some experiments to validate the results but they are not still available to insert in this publication.

6)      Can the authors assess other cell types involved in TME such as B cells, CAFs?

Thank you very much for your question. We have evaluated different immune cell types and, as you requested in question 2, we have added the cell populations evaluated by the panel.

7)      What is the predictive power and accuracy score of the 9-gene signature? Do all the genes contribute equally to the model?

Thank you very much for your question. We included in the analysis a nomogram, and the concordance index (C-index) derived from this indicated that our signature has good accuracy in predicting patient outcome. (paragraph 2.6, lines 133-141 of the result)

The assessment of the contribution of each gene to survival prediction is given by the regression models and by log-rank test score. The lower the value, the higher the contribution of a single gene will be.

8)      Can the authors perform any additional experiments to validate their 9-gene signature model?

Thank you very much for your suggestion. We are doing experiments to validate the signature by immunohistochemistry on the skin tissues of the discovery set. The next step is the validation on the international validation set and on a bigger group of patients in the future.

Minor points:

9)      In Figure 2A, add a scale bar and color key.

Thank you very much for your suggestion. We have modified the figure 2 as requested.

10)   Expand OS in line 36.

Thank you very much for your suggestion. We have improved the introduction, expanding the section on the OS correlated to the stage.

Round 2

Reviewer 3 Report

The authors have addressed most of the issues raised. Please make sure to note the limitations and future experiments in the discussion section before publication.